# The Emergence of Essential Sparsity in Large Pre-trained Models: The Weights that Matter

**Ajay Jaiswal[1], Shiwei Liu[1,2], Tianlong Chen[1], Zhangyang Wang[1]**
[1]University of Texas at Austin, [2]Eindhoven University of Technology
{ajayjaiswal, shiwei.liu, tianlong.chen, atlaswang}@utexas.edu

## Abstract

Large pre-trained transformers are *show-stealer* in modern-day deep learning, and it becomes crucial to comprehend the parsimonious patterns that exist within them as they grow in scale. With exploding parameter counts, Lottery Ticket Hypothesis (LTH) and its variants, have lost their pragmatism in sparsifying them due to high computation and memory bottleneck of repetitive *train-prune-retrain* routine of iterative magnitude pruning (IMP) which worsens with increasing model size. This paper comprehensively studies *induced sparse patterns* across multiple large pre-trained vision and language transformers. We propose the existence of – **"essential sparsity"** defined with a **sharp dropping point** beyond which the performance declines much faster w.r.t the rise of sparsity level, when we directly remove weights with the smallest magnitudes in **one-shot without re-training**. We also find essential sparsity to hold valid for N:M **sparsity patterns** as well as on **modern-scale large language models** (Vicuna-7B/13B). We also present an intriguing emerging phenomenon of **abrupt sparsification** during the pre-training of BERT, i.e., BERT suddenly becomes heavily sparse in pre-training after certain iterations. Moreover, our observations also indicate a **counter-intuitive** finding that BERT trained with a larger amount of pre-training data tends to have a better ability to condense knowledge in comparatively relatively fewer parameters. Lastly, we investigate the effect of the pre-training loss on essential sparsity and discover that self-supervised learning (SSL) objectives trigger stronger emergent sparsification properties than supervised learning (SL). Our codes are available at https://github.com/VITA-Group/essential_sparsity.

## 1 Introduction

Enormous increases in scale often permeate systems with unique new behavior. Transformers [1], swiftly after their introduction are scaling every day, dramatically improving the state-of-the-art performance on a wide array of real-world computer vision [2, 3, 4, 5, 6, 7], natural language processing [8, 9, 10, 11, 12, 13, 14, 15, 16] applications and leaderboards. With the astonishing explosion of parameter counts (millions to billions) in the past few years, while chasing performance gains, fine-tuning these large pre-trained models with non-industry standard hardware is becoming seemingly impossible, in addition to expensive inference and steep environmental cost. For instance, GPT-3 [17] contains 175 billion parameters and requires at least five 80GB A100 GPUs for inference [18].

In the hustle of building gigantic models, a parallel and growing field of model compression has been exploring the prospects to compress these gigantic models at the cost of marginal/no sacrifice in performance. Among many efforts for compressing models and accelerating inference, network pruning [19, 20, 21, 22, 23, 24, 25, 26], which removes the least important components from the model, varying from the low granularity (e.g., individual weights) to the high granularity (such as blocks, filters, and attention heads), stands out as one of the most effective techniques. Despite the enormous success of the Lottery Ticket Hypothesis (LTH) [27] and its variants for small-scale neural

37th Conference on Neural Information Processing Systems (NeurIPS 2023).

networks, there still exists a large **gap** in its practical adoption for large pre0trained transformers with billions of parameters because of the fully dense training routine of iterative magnitude pruning (IMP), which exacerbates with an increase in model capacity. Recently, many works [28, 29, 30, 20] are investigating strategies to make IMP pragmatic for large pre-trained models. However, they still rely on train-prune-retrain routine of IMP and downstream tasks to identify subnetworks. While chasing for gold-standard sparsity masks with computationally expensive approaches, it is critically important to understand pre-existing *high-quality sparse patterns* in these large transformers.

In our work, we reveal and study an intriguing, previously overlooked property of large pre-trained transformers – **"essential sparsity"**. In plain words (formal definition in Sec. 3), we define essential sparsity as ***sharp dropping point***, beyond which the fine-tuning performance after one-shot pruning declines much faster w.r.t. the sparsity level rise. We directly remove weights with the smallest magnitudes in **one-shot** from the pre-trained checkpoints, thereby the identified subnetwork mask requires no extra computational overhead to spot and remains identical across all downstream tasks.

Essential sparsity is an important yet understudied induced property of large pre-trained transformers, indicating that at any time, *a significant proportion of the weights in them can be removed **free** of any calibration data or post-optimization, although the proportion may vary depending on the complexity of the downstream task*. One important practical implication conveyed by this observation, as overlooked by prior work, is summarized as: *within the sparsity range induced by "essential sparsity", the simplest possible pruning technique as aforementioned performs the same well as any fancy technique such as LTH, and even their identified sparse masks are highly similar.*

Our work, for the **first** time, conducts a comprehensive study of the existence of essential sparsity across multiple vision and language transformers with varying scales and training strategies. Besides, we observe essential sparsity to **hold valid for various** `N:M` **sparsity patterns** with hardware speedup potentials. In addition, our experiments using the popular `MMLU` benchmark [31] on `Vicuna-7B/13B` illustrate essential sparsity observations remain **true for modern-scale large language models (LLMs)**, indicating the existence of high-quality sparse subnetworks within dense pre-trained checkpoint.

Next, we study the emergence of those sparsification properties during the pre-training dynamics, using BERT as the focused subject of study. We found an intriguing phenomenon of **abrupt sparsification**, i.e., BERT suddenly becomes heavily sparse after certain number of training iterations. As we vary the pre-training dataset size. our observation indicates another **counter-intuitive** finding that BERT trained with a *larger amount of pre-training data tend to have a better ability to condense knowledge in relatively fewer parameters*. We also dive deep into the effect of the pre-training loss on essential sparsity and discover that self-supervised learning (SSL) objectives trigger stronger emergent sparsification properties than supervised learning (SL).

Key contributions for our work can be summarized as:

- We found the **ubiquitous existence of essential sparsity** across large pre-trained transformer models of varying scales for vision and language, irrespective of the training strategy used for pre-training them. High-quality sparse masks comparable to lottery tickets [27] within the essential sparsity range can be spotted *free of inference or post-optimization* by merely selecting the lowest magnitude weights, that requires no expensive repetitive train-prune-retrain routine. The observation holds true for both unstructured and `N:M` sparsity patterns.

- Our comparison of the sparse masks obtained by selecting the lowest magnitude weights with the lottery tickets within the essential sparsity range, surprisingly unveils a **notably high cosine similarity** (>98%) across various downstream tasks from NLP and CV. This observation sends a strong message that in large pre-trained models, LTH perhaps enjoys little additional privilege despite utilizing enormous computational costs.

- While studying the emergence of sparsification properties during the pre-training dynamics in BERTs, we found an intriguing phenomenon of **abrupt sparsification**, i.e., BERT suddenly becomes heavily sparse after certain number of training iterations. We additionally found a **counter-intuitive** observation that BERT pre-trained with larger amount of data tends to be more sparser, i.e., they become better at *knowledge abstraction with fewer parameters*.

- We additionally study the effect of the pre-training loss on the emerging sparsity of FMs. When we switch between supervised learning (SL) and self-supervised learning (SSL) objectives on ViT, we observed that **SSL tends to have better emergent sparsification**

properties, thereby more friendly to pruning. We further provide layer-wise visualization to understand what the sparsity learned by SSL vs SL looks like.

## 2   Related Work

**Sparse Neural Networks (SNNs).**    Sparsity in deep neural networks is usually introduced by model compression [32, 33] which removes the redundant parameters. Based on the type of sparsity patterns, it can be categorized into two families: ($i$) *unstructured sparsity* [33, 34, 35] where the non-zero elements are irregularly distributed; ($ii$) *structured sparsity* [36, 37, 38] where a group of parameters is eliminated like a convolutional kernel in convolutional neural networks or an attention head in the transformer. In general, the former sparsity pattern obtains a better performance thanks to its flexibility, while the latter sparse pattern tends to be more hardware friendly. Many important SNN works start by studying the former and then turn to the latter as a special case.

Meantime, according to the timing to perform dimensionality reduction, sparsity can be obtained in the post-training, during-training, and prior-training of deep neural networks. ($i$) *Post-Training*. To pursue inference time efficiency, trained models can be pruned dramatically with marginal loss of performance with respect to certain heuristics, including zero-order [33], first-order [39, 40, 41], and second-order [32, 42, 43] criteria. Among these algorithms, the weight magnitude-based approach (*e.g.* iterative magnitude pruning) is a popular option. Especially, it is frequently adopted by the Lottery Ticket Hypothesis [19] to produce sparse NNs with undamaged trainability and expressiveness. ($ii$) *During-Training*. On the contrary to pruning a well-trained model for inference efficiency, during-training sparsification [44] also enjoys computational savings for model training. It normally first optimizes a dense network for several iterations, then gradually sparsifies it with a pre-defined schedule, and finally leads to lightweight sparse NNs. For example, [45, 46, 47] leverage $\ell_0$ or $\ell_1$ penalties to encourage weight sparsity during the network training, hoping to zero out unimportant parameters. [48, 49] cast it as an optimization problem with reparameterization and bi-level forms, respectively. Another fashion is dynamic sparsity exploration [50, 51, 52, 53, 54, 55, 56, 57, 58, 59] which allows pruned connections can be re-activated in the latter training stage. ($iii$) *Prior-Training*. Identifying the critical sparsity patterns at the initialization is one exciting rising sub-field. It determines the sparse connectivities in a very early stage, like one iteration [25, 60] or a few iterations [61, 62]. In this paper, we mainly investigate post-training unstructured SNNs.

**Sparsity in Pre-Trained Transformers.**    Pre-trained Transformers have become *de facto* choice for numerous applications of natural language processing (NLP) [8, 9, 10, 15, 16] and computer vision [2, 7, 63]. Their impressive performance is partially credited to their tremendous learning capacity empowered by huge amounts of model parameters. Unfortunately, such successes come with burning thousands of GPUs/TPUs for thousands of hours [64, 17], even for a single round of model training. To address this resource-intensive concern and improve the affordability of these transformers, plenty of pioneering researchers devote themselves in this area [40, 20, 30, 65, 66, 67, 68, 69, 70]. Rather than proposing a new sparsification method, this paper reveals the blessings of induced essential sparsity during the pre-training and how we can capitalize it to prune large pre-trained models without any computational overhead.

## 3   Experimental Settings

**Network and Pre-Trained Transformers.**    We consider {BERT$_{\text{Base}}$ [64], BERT$_{\text{Large}}$ [64], OPT$_{\text{125M}}$ [71], OPT$_{\text{350M}}$ [71], and OPT$_{\text{1.3B}}$ [71]} and {ViT$_{\text{Base}}$ [2], ViT$_{\text{Large}}$ [2], and DiNO$_{\text{Base}}$ [63] for NLP and CV applications respectively. Their officially pre-trained models[1] are adopted in our experiments. To be specific, let $f(x; \theta_p)$ be the output of a transformer with pre-trained model parameters $\theta_p$ and input sample $x$.

**Datasets, Training, and Evaluation.**    For downstream tasks in NLP, we consider {MNLI, QNLI, QQP, SST-2} from GLUE [13], RTE from SuperGLUE [72], and SQuAD v1.1 [73]. As for vision downstream applications, we examine {CIFAR-10, CIFAR-100, Tiny-ImageNet [74]}. We also use the Arithmetic Reasoning task from the recently proposed SMC-Benchmark [75] and consider popular math word problem datasets: (1) the widely used MAWPS benchmark [76] composed of 2,373 problems; (2) the arithmetic subset of ASDiv-A [77] - with 1,218 math problems; (3) the more

---

[1]Check supplementary materials for pre-trained model details.



Figure 1: Naturally induced sparsity patterns of `bert-base-uncased` across the components of transformer blocks. The pre-trained model is pruned by 21.50% using one-shot-magnitude pruning. Yellow dots indicate the location of pruned low-magnitude weights.

Table 1: Downstream tasks fine-tuning details. Learning rate decay linearly from initial value to 0.

| Settings | Natural Language Processing | | | | | | Computer Vision | | | |
|---|---|---|---|---|---|---|---|---|---|---|
| | MNLI | QNLI | QQP | RTE | SST-2 | SQuAD v1.1 | CIFAR-10 | CIFAR-100 | Fashion-MNIST | Tiny-ImageNet |
| # Training Ex | 392,704 | 104,768 | 363,872 | 2,496 | 67,360 | 88,656 | 45,000 | 45,000 | 55,000 | 90,000 |
| # Epoch | 3 | 4 | 3 | 5 | 5 | 3 | 8 | 8 | 8 | 5 |
| Batch Size | 32 | 32 | 32 | 32 | 32 | 16 | 64 | 64 | 64 | 64 |
| Learning Rate | $2e-5$ | $2e-5$ | $2e-5$ | $2e-5$ | $2e-5$ | $3e-5$ | $2e-5$ | $2e-5$ | $2e-5$ | $2e-5$ |
| Optimizer | AdamW with decay $(\alpha) = 1 \times 10^{-8}$ | | | | | | AdamW with decay $(\alpha) = 2 \times 10^{-8}$ | | | |
| Eval. Metric | Matched Acc. | Accuracy | | | | F1-score | Accuracy (Top-1) | | | |

challenging SVAMP [78] dataset which is created by applying complex types of variations to the samples from ASDiv-A. The task difficulty monotonically increases from MAWPS to ASDiv-A, and to SVAMP. We adopt the default dataset split for training and evaluation for our downstream application. More detailed configurations are collected in Table 1. For SMC-benchmark, we strictly followed the settings proposed in the official implementation[2]

**Sparse Neural Networks (SNNs).** The weights of a SNN can be depicted as $m \odot \theta$, where $m \in \{0,1\}^{|\theta|}$ is a binary mask with the same dimensionality as $\theta$ and $\odot$ is the element-wise product. Let $\mathcal{E}^{\mathcal{T}}(f(x;\theta))$ denotes the evaluation function of model outputs $f(x;\theta)$ on the corresponding task $\mathcal{T}$ (which might involve fine-tuning). $\mathcal{P}_\rho(\cdot)$ is the sparsification algorithm which turns a portion $\rho$ of "1" elements in the sparse mask $m$ into "0"s. Here is a formal definition of **Essential Sparsity** below.

> **Essential Sparsity**. If $\mathcal{E}^{\mathcal{T}}(f(x;m \odot \theta)) \geq \mathcal{E}^{\mathcal{T}}(f(x;\theta)) - \epsilon$, and $\mathcal{E}^{\mathcal{T}}(f(x;\mathcal{P}_\rho(m) \odot \theta)) < \mathcal{E}^{\mathcal{T}}(f(x;\theta)) - \epsilon$ where the value of $\rho$ and $\epsilon$ are small. Then, the according sparsity $1 - \frac{\|m\|_0}{|m|}$ is named as Essential Sparsity for the model $f$ on task $\mathcal{T}$.

As detailed above, the model at the essential sparsity usually has a turning point performance, which means further pruning even a small portion $\rho$ of weights leads to at least $\epsilon$ performance drop, compared to its dense counterpart $\mathcal{E}^{\mathcal{T}}(f(x,\theta))$. In our case, $\epsilon$ is set as 1%. In plain language, the turning point of essential sparsity defines a *sparsity range* with two characteristics: (i) *within this range*, the one-shot pruned model performs as well as or better than dense baseline, without re-training; (ii) *beyond this range*, a notable accuracy drop becomes observable after one-shot magnitude pruning.

**Pruning Methods.** To find the desired sparse mask $m$, we use two classic weight magnitude pruning algorithms [33, 19]. *One-shot Magnitude Pruning* (OMP): we directly eliminate a pre-defined portion of parameters from $\theta_p$ with the least absolute magnitude. `Lottery Tickets Pruning` (LTP): $(i)$ we first train the model to converge on a downstream task $\mathcal{T}$; $(ii)$ then remove a portion of the smallest weights and reset the remaining weight to their initialized value from pre-training $\theta_p$; $(iii)$ such processes will be repeated until reaching the desired sparsity.

Two crucial facts are noteworthy. First, starting from the pre-trained model, unlike LTP which seeks downstream-specific masks with additional training, the OMP sparse masks require **no additional training** to identify and are **agnostic to downstream applications**. Since they are estimated directly from the pre-trained weight checkpoint, the OMP sparse mask remains the same for all downstream applications (we only continue to fine-tune the non-zero weights with the same mask, over different downstream datasets). Second, for the same above reason, a higher-sparsity OMP mask is naturally **nested** within a lower-sparsity mask, as long as they are pruned from the same pre-trained checkpoint using OMP. The same nested property does not hold for LTP-obtained masks.

---

[2]`SMC-Benchmark` fine-tuning setting details: Arithmetic Reasoning

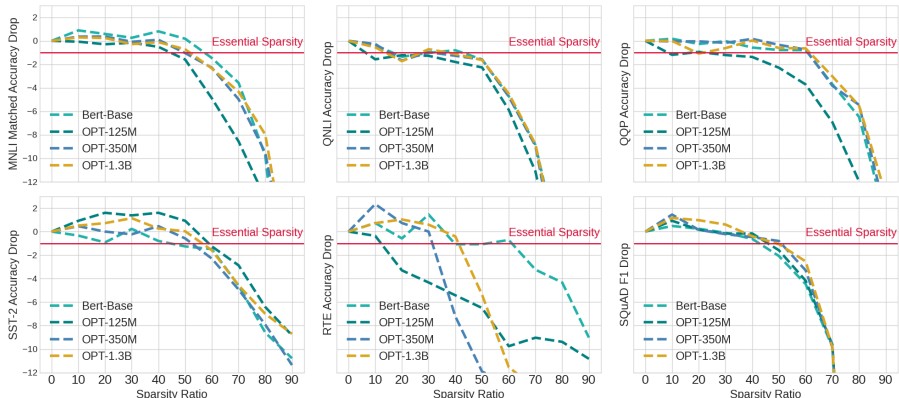

Figure 2: Fine-tuning performance drop estimated with respect to dense counterpart for various downstream tasks of NLP pre-trained models (`bert-base, OPT-125m, OPT-350m, OPT-1.3B`). Note that for fair evaluation, we have used exactly same fine-tuning settings across all pruning ratios.

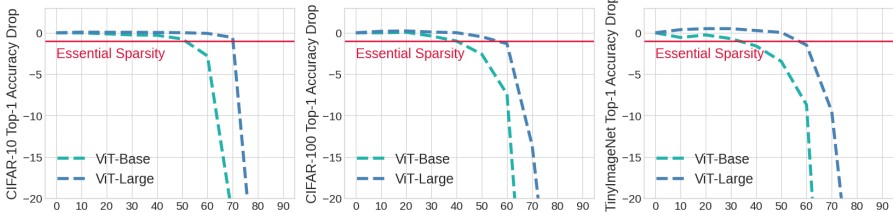

Figure 3: Fine-tuning performance drop estimated with respect to dense counterpart for various downstream tasks of CV pre-trained models (`ViT-base` & `ViT-large`).

## 4 Essential Sparsity Exists in Pre-trained Language and Vision Models

**Revisiting sparsity in Pre-trained Transformers:** The unconstrained growth in parameters has resulted in significant computational costs and excessive memory demands. The trend undoubtedly continues with transformers on the forefront, where more and more layers are stacked with dense attention blocks (eg. GPT-3 has surprisingly 175 billion parameters) necessitating substantial computational resources and extended training or fine-tuning durations. Unlike extensive efforts explored in the past for pruning ResNets and related families of networks, pruning large-scale pre-trained transformer models hasn't received enough attention.

Lottery Ticket Hypothesis (LTH) based approaches have been recently explored to prune BERT-base size pre-trained models. However, they lose their pragmatism due to the expensive dense train-prune-retrain routine of IMP and non-transferable characteristics of identified subnetworks across similar tasks [20]. oBERT [66] proposes second-order to BERT-level scale by allowing for pruning blocks of weights. Note that, although these methods provide interesting approaches to prune BERT scale transformers, they *fail to sufficiently recognize the strength of pre-existing, more easily accessible high-quality sparse patterns* in pre-trained large models invariant of the scale, training data modality, and strategy. This paper's primary goal is to bring the sparse community's attention towards the strength of ubiquitously induced sparse patterns during the pre-training of large transformers and encourage them to effectively capitalize it during the design of more sophisticated pruning algorithms.

**Essential Sparsity:** As defined in Section 3, essential sparsity indicates the presence of naturally induced sparse structures in large pre-trained models with respect to the lowest magnitude weights. Figure 1 represents the induced sparsity distribution of `bert-base-uncased` pre-trained model with 21.50% close to zero weights, which we have been able to remove without **ANY** drop of performance across any of our evaluation downstream datasets. This sought a strong message about the existence of free, universal, and available to consume sparse mask $(m \cdot \theta_p)$ without any computational overhead induced by sophisticated pruning algorithms. In addition, a closer observation of the distributed sparsity ratios across different modules of the transformer block indicates that the majority of the low-magnitude weights are concentrated in the dense feed-forward layers. Such findings conform

with the success of Mixture-of-Experts [79] and provide cues for future development of pruning algorithms to exploit the over-parameterization of dense feed-forward layers.

We now enlist our key findings related to essential sparsity in large pre-trained transformer models:

- In all vision and language models, we find the existence of essential sparsity and the sharp turning point of the sparsity-performance curve.

- The sharp turning point of essential sparsity is downstream task-dependent and can vary depending on the task complexity.

- The sharpness behavior of essential sparsity is more profound in vision pre-trained models in comparison with language models.

- Essential sparsity holds valid for recently proposed pruning benchmark SMC-bench [75].

- Essential sparsity holds valid for `N:M` structured sparsity patterns [80] with potential speedup.

- Self-supervised learning objective triggers stronger emergent sparsification properties than supervised learning in models having exactly the same architecture.

**Universal Existence of Essential Sparsity:** We comprehensively evaluate the extent to which essential sparsity exists in the large pre-trained language models (CV and NLP) with a standard pre-trained initialization $\theta_p$. For NLP, we studied `bert-base`, `OPT-125M`, `OPT-350M`, `OPT-1.3B` and used `MNLI`, `QNLI`, `QQP`, `SST-2`, `RTE, and SQuAD1.1` to estimate the effect the removing low-magnitude weights on downstream performance. On the other hand, for CV, we rely on `ViT-Base` and `ViT-Large` models and use `CIFAR-10`, `CIFAR-100`, `TinyImageNet` datasets. Figure 2 illustrate the performance drop (y-axis) of various NLP downstream task fine-tuning with mask $(m \cdot \theta_p^{x\%})$, when we remove $x\%$ of lowest magnitude weights. Similarly,

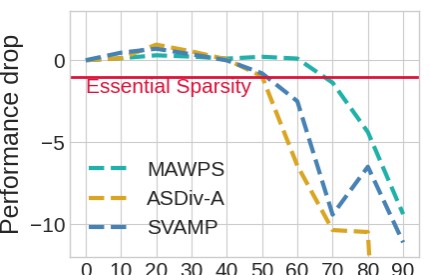

Figure 4: Fine-tuning performance drop of `bert-base` on Arithmetic Reasoning datasets in the SMC-benchmark [75].

Figure 3 illustrates the effect of pruning $x\%$ low-magnitude weights on the fine-tuning performance of CV downstream tasks. Moreover, Figure 4 presents the existence of essential sparsity for a recently proposed sparsity benchmark, named SMC-Bench [75]. It is interesting to observe that essential sparsity exists in all CV and NLP pre-trained models, and $\sim 30 - 50\%$ of weights can be removed at free without any significant drop in performance. Note that these masks are identified prior to fine-tuning, on the pre-trained weights and thereby remain the same for all the downstream tasks, indicating no requirement for bookkeeping LTH-type task-specific masks. It is also important to appreciate the significance of removing $\sim 40\%$ (which translates to > 500 million parameters) of OPT-1.3B at no cost without any significant performance drop.

**Eesential Sparsity for Structured `N:M` Sparse Patterns:** Considering the demand for practical speedup, we also conduct evaluations to understand the essential sparsity for the hardware-friendly structured `N:M` sparsity. A neural network with `N:M` sparsity satisfies that, in each group of $M$ consecutive weights of the network, there are at most $N$ weights have non-zero values. We studied `OPT-350M` and used `MNLI`, `QNLI`, `RTE`, `SST-2` to estimate the effect of removing low-magnitude weights in structured `N:M` fashion following [80]. Figure 5 illustrates the performance drop (y-axis) of `OPT-350M` on

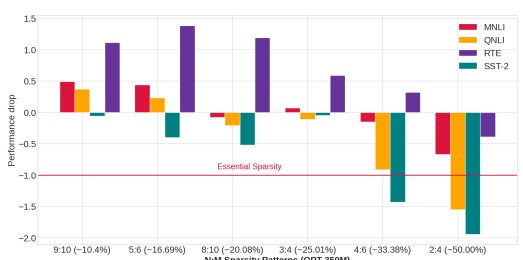

Figure 5: Fine-tuning performance drop of `OPT-350M` at GLUE dataset wrt. dense counterpart on multiple `N:M` sparsity patterns [80] masks.

various downstream datasets with multiple `N:M` sparse mask $(m \cdot \theta_p^{x\%})$. It can be in general observed essential sparsity holds valid for structured `N:M` sparsity patterns, and a large fraction of low-magnitude weights can be removed at free without significant drop in downstream performance.

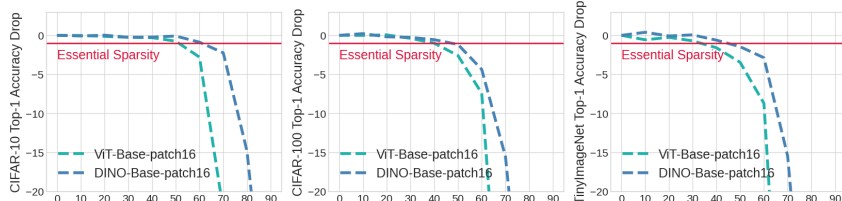

Figure 6: Essential Sparsity and performance comparison `ViT-base` and `DINO-base` which share the same architecture but pre-trained using supervised (SL) and self-supervised learning (SSL) objectives. It can be observed that the SSL induces a better sparsification ability in the pre-trained checkpoint.

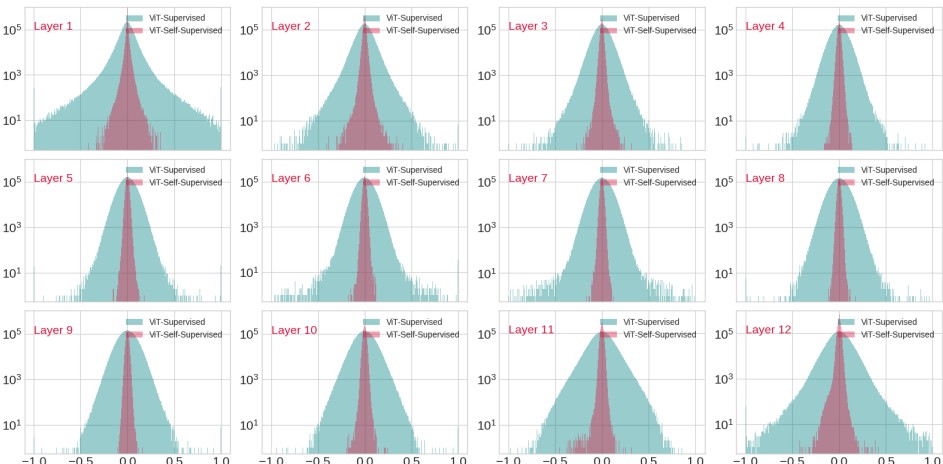

Figure 7: Layer-wise weight distribution of `ViT-base` and `DINO-base` trained using supervised and self-supervised learning objective. Note that the weights of both pre-trained models are normalized using `sklearn` for fair comparison. Additionally, `DINO` has 14.37% more zero weights than ViT.

**Essential Sparsity and Sharp Turning Behaviour:** In this section, we attempt to bring attention towards the sharp turning point behavior of essential sparsity. Across all our vision and language models and related downstream tasks (Figure 2, 3, & 4), we observed that after a certain removal of a certain proportion of low-magnitude weights, the downstream fine-tuning ability of pre-trained models **sharply** drops. This is a clear indication that the pre-training knowledge resides in a fraction of high-magnitude weight regions and if we our one-shot pruning touches that region, it non-linearly impacts the transfer learning ability of the pre-trained checkpoint. Our experiments reveal that this *sharp-turning behavior is not dependent on model size but on the downstream task*. Larger model doesn't implicate that it can be pruned for a higher proportion of low-magnitude weight, without observing the sharp drop. For example, `bert-base` observes the sharp turning point at around 60% sparsity while OPT-1.3B can not be pruned beyond 40% without observing the sharp performance drop on `RTE` task, although it has $\sim 10$ times more parameters than `bert-base`. Also, it is interesting to observe that although `OPT-125M` and `bert-base` have similar performance count, `bert-base` illustrate more friendly behavior to pruning, and can be pruned to comparatively higher sparsity ratio than `OPT-125M` on all our evaluation downstream tasks.

**Influence of Supervised versus Self-supervised Learning Objectives:** With the recent success of self-supervised learning (SSL) in pre-training large transformer models and its ability to scale to enormous internet-size data, it is interesting to understand how SSL learning objectives favor sparsity. Due to the unavailability of supervised datasets to pre-train BERT-scale models, we switch to Vision transformers, and use `ViT-base` and its self-supervised version `DINO-base` which inherit exactly the same architecture. Figure 6 illustrates fine-tuning performance drop of `ViT-base` and `DINO` with an increase in the proportion of low-magnitude weight removal from pre-trained checkpoints. Across all tasks, fine-tuning performance doesn't suffer any drop till $30 - 40\%$ weights are removed.

One interesting observation is that `DINO-base` tends to be more robust to low-weight removal, and has comparatively better essential sparsity across all tasks. To further investigate why we can prune

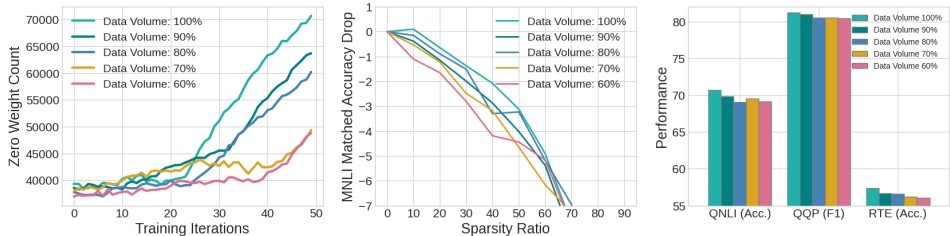

Figure 8: Plot Description in order left-right. (i) Zero-weight count of 5 pre-training experiments of `Bert-base` using `bookcorpus` dataset from HuggingFace with varying percentages of randomly selected data volume with exactly the same pre-training setting. (ii) Downstream performance of the pre-trained `Bert-base` models with varying data volume across different sparsity ratios on `MNLI`. (iii) Downstream performance of 5 dense pre-trained models on `QNLI, QQP, RTE`.

`DINO-base` more than `ViT-base`, we examine the weight magnitude distribution across the layers of the transformer blocks from the pre-trained checkpoints. Figure 7 presents the normalized layer-wise weight distribution of `DINO-base` and `ViT-base`. It can be clearly observed that SSL-based `DINO` tends to have a weight distribution more friendly to pruning with a significantly large amount of weights concentrated around zero magnitudes. More concretely, we found that `DINO-base` have $\sim 14\%$ more zero weights than `ViT-base`, which justifies its higher essential sparsity.

## 5 How Essential Sparsity Emerges during the Pre-Training Dynamics

Scaling the volume of pre-training data volume is widely believed to favor transfer performance in many downstream tasks in a desirable way [81]. Although, this relationship has been extensively studied recently in many works [81, 82, 83, 84], pre-training data volume role in inducing sparse properties in the large transformers is still **unexplored**. In this section, we ask an important question: *How does the volume of pre-training data impact the emerging sparse patterns in large transformers, and if it improvises their prunability?*

To this end, we designed custom experiments to pre-train `bert-base` from scratch using HuggingFace `bookcorpus` with a vocabulary size of 30,522. We created 5 different pre-training datasets by randomly selecting $100\%, 90\%, 80\%, 70\%, 60\%$ of the training samples from `bookcorpus` and pre-train for 50k iteration each to ensure that all models receive the same amount of gradients. Note that we maintain exactly same training settings for all models to retain fair comparison and save the pre-training checkpoints every 1000 iterations. We now enlist our key findings related to pre-training data volume and induced sparse patterns in transformers:

- We observe an interesting new phenomenon of **abrupt sparsification**, i.e., the introduction of sudden high sparsity, during pre-training `bert-base` models, regardless of data size.

- We additionally observed a counter-intuitive finding that `bert-base` trained with a larger amount of pre-training data tends to have better emergence of induced sparsity.

- Across all sparsity level, we found `bert-base` trained with a larger volume of training data, enjoys better prunability, and achieve better performance on the downstream task (`MNLI`).

Figure 8(i) illustrate the emergence of sparse patterns during pre-training of `bert-base` with different pre-training data volume. We plotted the number of zero weights that emerged in the pre-training checkpoints every 1k iterations. It can be clearly observed that in all settings, the number of zero weights suddenly grow up, indicating the models start abstracting pre-training data knowledge into fewer parameter set. For instance, we find this sharp turn around 22-25k iterations for $100\%, 90\%, 80\%$ settings while around 40k iterations for $70\%, 60\%$ settings. In addition, we found that `bert` trained with a larger amount of pre-training data tends to have better emergence of induced sparsity. We argue that `bert-base` tend to learn more generalizable features and develop the capability to abstract knowledge To further investigate how this sparsity emergence further influences the prunability of the pre-trained models, we examined their performance across multiple sparsity ratios (Figure 8(ii)) on `MNLI` and found it to be in harmony with the magnitude of induced sparsity, i.e., models with high induced sparsity patterns during pre-training tend to perform better when we remove the existing low magnitude weights and fine-tune them on downstream tasks. In dense

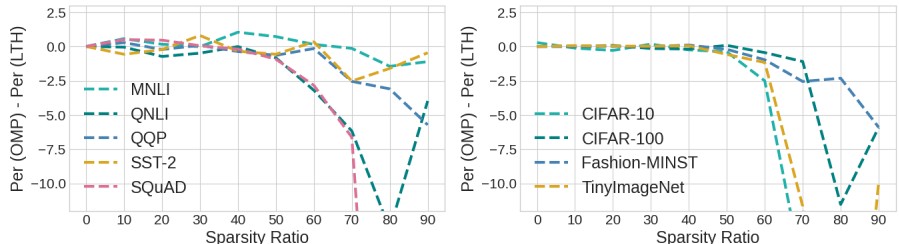

Figure 9: Performance difference comparison of fine-tuning of masks identified by LTH and OMP from `bert-base` (left) and `ViT-base` (right) across multiple downstream tasks.

settings on `QNLI, QQP, RTE` (Figure 8(iii)), we found that an increase in pre-training data volume has favorable benefits on the downstream performance with similar fine-tuning hyperparameters.

# 6 Essential Sparsity and Lottery Ticket Hypothesis

Lottery Ticket Hypothesis deviates from the convention of after-training pruning, and points to the existence of independently trainable sparse subnetworks from scratch that can match the performance of dense networks. Since its introduction, it has been very successful and widely adopted in compressing small-scale (eg. ResNet family) models. However, it is significantly limited in the practical adoption for large pre-trained models due to *train-prune-retrain* routine of IMP. In this section, we ask: *How effective LTH is within the essential sparsity range, and does it bring any additional privilege which can substantiate its computational cost?* Our key takeaway can be summarized as:

- Within the essential sparsity range of `bert-base` and `ViT-base`, we do not find any significant fine-tuning performance difference of the mask identified by LTH and one by removing lowest magnitude weights (OMP) across multiple downstream tasks.

- We surprisingly found the existence of high cosine similarity($> 96\%$) across the masks identified by LTH and OMP within the essential sparsity range.

- Mask similarity between LTH and OMP decreases with an increase in the sparsity ratio. This corroborates with the benefit of LTH in the high sparsity range, where LTH is able to identify task-fitting masks to retain performance due to repetitive prune and retrain routine.

Figure 9 illustrates the performance difference comparison of fine-tuning of masks identified by LTH and OMP from `bert-base` and `ViT-base` across multiple downstream tasks. It can be clearly observed that for sparsity ratio below $50\%$, we do not find any *significant difference in performance between expensive LTH and free-of-cost OMP pruning within the essential sparsity range*. A deeper analysis (Figure 10) across the masks from LTH and OMP unveils a surprising observation about the existence of significantly high cosine similarity across. This observation supports the findings in Figure 9 about the matching performance of LTH and OMP and convey a strong and impressive message that with the essential sparsity range, LTH doesn't provide any additional privilege. However, it is important to note that LTH is very effective in the high sparsity range beyond essential sparsity, due to its ability to find task-fitting mask using train-prune-retrain procedure but they tend to be non-transferable across different downstream tasks [20]. Moreover, it can be observed from Figure 10, even in the essential sparsity range, mask similarity across tasks by LTH is comparatively low than OMP based masks due to the strong intertwine of LTH with the downstream tasks.

# 7 Scaling Essential Sparsity to Modern-Scale LLMs: A Case Study

Large Language Models (LLMs) recently reshape the field of NLP with remarkable performance benefits across a range of complex language benchmarks. However, due to their gigantic size and computational costs; they still remain out of reach for many researchers and small industries. In this section, we investigate the presence of essential sparsity in two popular LLM (`Vicuna-7B & 13B`) with one-shot magnitude pruning. Note that unlike our previous setting, where we perform sparse fine-tuning after the mask is identified; here we do not perform downstream task-specific fine-tuning and evaluate the performance directly "zero-shot" on the sparsified pre-trained checkpoint.

Figure 11(a) illustrates the performance drop of `Vicuna-7B` and `Vicuna-13B` on popular `MMLU` benchmark (Stem, Humanities, Social Science, Others) when $x\%$ of the lowest magnitude weights

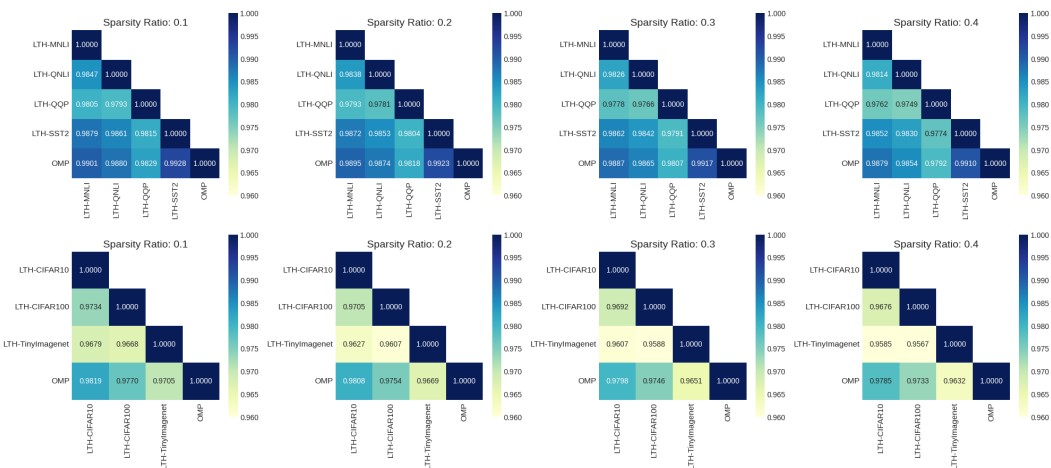

Figure 10: Cosine similarity between the masks obtained by LTH (depending on downstream task) and OMP on `bert-base` (Row 1) and `ViT-base` (Row 2) for sparsity ratio $s \in \{10\%, 20\%, 30\%, 40\%\}$. High cosine similarity indicate masks identified by LTH and OMP are significantly similar.

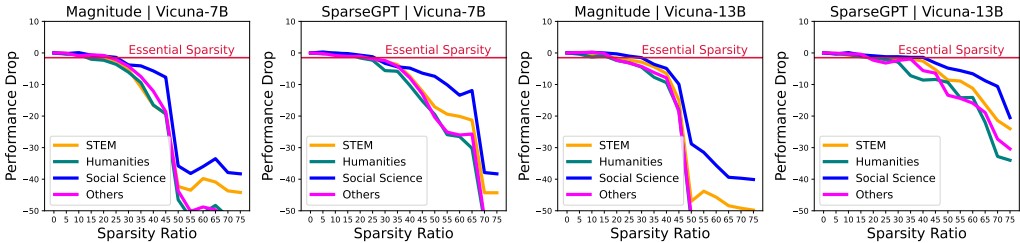

Figure 11: Performance drop of `Vicuna-7B` and 13B on `MMLU` benchmark [31] w.r.t. the dense counterpart, using OMP and the recently proposed SparseGPT [85]. This indicates a notable fraction of weights of `Vicuna-7B/13B` can also be removed at free without any significant drop in performance.

are removed from the dense pre-trained checkpoint. It is interesting to see that our essential sparsity observations hold true even for modern LLMs, sending a favorable signal about the hidden existence of high-quality sparse subnetwork which can be identified free of calibration data or post-optimization, in dense pre-trained checkpoints. To further enrich our study, we replaced OMP with the recently proposed SparseGPT [85] and found it to have generally consistent trends with OMP (Figure 11(b)).

Compared to SparseGPT, our research uncovers the surprising simplicity of LLM pruning *within a specific sparsity range*. The remarkable effectiveness of one-shot, magnitude-based pruning not only establishes a robust baseline but also suggests considerable practical value, for instance, in enabling economical "on-the-fly" LLM pruning that adapts to fluctuating resource availability during testing. Nonetheless, it's worth noting that more refined pruning strategies like SparseGPT can further push the boundary of essential sparsity and identify better sparse subnetworks at comparatively higher sparsity ratios, albeit at relatively higher expenses. Bridging the performance divide between OMP and SparseGPT remains an intriguing avenue for future investigation.

## 8 Conclusion

We comprehensively study induced sparse patterns across large pre-trained vision and language transformers. We experimentally validated the ubiquitous existence of essential sparsity across large pre-trained transformer models of varying scales for vision and language (including LLMs), irrespective of the training strategy used for pre-training them. We also present an intriguing emerging phenomenon of abrupt sparsification during the pre-training of transformers and its ability to abstract knowledge within few parameter subset with increasing pre-training data volume. Lastly, we studied the performance of LTH with respect to essential sparsity. Our future work will aim to extend our essential sparsity observations to more gigantic models, and to push for higher sparsity ranges.

## Acknowledgement

The research is in part supported by the Intelligence Advanced Research Projects Activity (IARPA) under Contract No. 2022-21102100004. The views and conclusions contained herein are those of the authors and should not be interpreted as necessarily representing the official policies, either expressed or implied, of IARPA or the U.S. Government. The U.S. Government is authorized to reproduce and distribute reprints for governmental purposes notwithstanding any copyright annotation therein.

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
