# The Emergence of Essential Sparsity in Large Pre-trained Models: The Weights that Matter

## 1 Supplementary Material

### 1.1 Fine-tuning Details of various Computer Vision and NLP tasks.

Table 1: Downstream tasks fine-tuning details. Learning rate decay linearly from initial value to 0.

| Settings | Natural Language Processing | | | | | | Computer Vision | | | |
|---|---|---|---|---|---|---|---|---|---|---|
| | MNLI | QNLI | QQP | RTE | SST-2 | SQuAD v1.1 | CIFAR-10 | CIFAR-100 | Fashion-MNIST | Tiny-ImageNet |
| # Training Ex | 392,704 | 104,768 | 363,872 | 2,496 | 67,360 | 88,656 | 45,000 | 45,000 | 55,000 | 90,000 |
| # Epoch | 3 | 4 | 3 | 5 | 5 | 3 | 8 | 8 | 8 | 5 |
| Batch Size | 32 | 32 | 32 | 32 | 32 | 16 | 64 | 64 | 64 | 64 |
| Learning Rate | $2e-5$ | $2e-5$ | $2e-5$ | $2e-5$ | $2e-5$ | $3e-5$ | $2e-5$ | $2e-5$ | $2e-5$ | $2e-5$ |
| Optimizer | AdamW with decay ($\alpha$) = $1 \times 10^{-8}$ | | | | | | AdamW with decay ($\alpha$) = $2 \times 10^{-8}$ | | | |
| Eval. Metric | Matched Acc. | Accuracy | | | | F1-score | Accuracy (Top-1) | | | |

### 1.2 SMC-Bench Arithmetic reasoning Task Settings

Table 2: Hyperparameters and training configurations used for models on Arithmetic Reasoning.

| Datasets | MAVPS, ASDiv-A, SVAMP |
|---|---|
| Pre-trained Embeddings | `bert-base` |
| Embedding Size | [768] |
| Hidden Size | [384] |
| Number of Layers | [2] |
| Learning Rate | [8e-4] |
| Weight Decay | [1e-5] |
| Embedding LR | [1e-5] |
| Batch Size | [4 (MAVPS, ASDiv-A), 8 (SVAMP)] |
| Dropout | [0.5] |
| Adam | [1e-08] |
| Adam $\beta_1$ | [0.9] |
| Adam$\beta_2$ | [0.999] |
| Training time | 50 epochs |

### 1.3 Pre-trained Compter Vision and NLP Model Details

Table 3: Download links for various Pre-trained NLP and Vision Models.

| Model Name | Download Link |
|---|---|
| bert-base | https://huggingface.co/bert-base-uncased |
| bert-large | https://huggingface.co/bert-large-uncased |
| OPT-125M | https://huggingface.co/facebook/opt-125m |
| OPT-350M | https://huggingface.co/facebook/opt-350m |
| OPT-1.3B | https://huggingface.co/facebook/opt-1.3b |
| ViT-base | https://huggingface.co/timm |
| ViT-large | https://huggingface.co/timm |
| DINO-base | https://github.com/facebookresearch/dino |
| DINO-large | https://github.com/facebookresearch/dino |