# OpenReview forum: "The Emergence of Essential Sparsity in Large Pre-trained Models: The Weights that Matter"
_NeurIPS.cc/2023/Conference — NeurIPS 2023 poster_

### Official Review · Reviewer_dYgU · 2023-07-05

**Soundness:** 3 good
**Presentation:** 4 excellent
**Contribution:** 4 excellent
**Rating:** 8
**Confidence:** 5

**Summary:**

This paper presents a comprehensive study on the induced sparse patterns across multiple large pre-trained vision and language transformers. The authors propose the existence of "essential sparsity" and present intriguing findings on abrupt sparsification during the pre-training of transformers and the effect of pre-training data on knowledge condensation.

**Strengths:**

The main strength of this paper is that it validates the ubiquitous existence of essential sparsity across large pre-trained transformer models of varying scales for vision and language, irrespective of the training strategy used for pre-training them.

The authors used various datasets and models to validate the existence of "essential sparsity" and presented intriguing findings on abrupt sparsification during the pre-training of transformers and the effect of pre-training data on knowledge condensation. The authors also compared the performance of models with and without sparsification and found that one-shot sparsification without re-training does not significantly affect the performance of downstream tasks. In Section 6, the authors further carefully analyzed the connection between LTH and essential sparsity, and showed the former to become potentially less necessary or relevant in larger models. Those are valuable insights of broad interest to the sparsity research community.

Based on the methodology and results presented in the paper, the experiment presented is sound enough to support the idea of the existence of essential sparsity in large pre-trained models (bert-base, OPT-125m, OPT-350m, OPT-1.3B). In particular, the authors present several quite surprising findings related to the existence of essential sparsity in large pre-trained transformer models.
-	Firstly, the authors found that BERT suddenly becomes heavily sparse after a certain number of training iterations, which is unobserved before and not well-understood. This finding suggests that there may be underlying “phase transition”-like mechanisms in the pre-training dynamics of transformers that are responsible for inducing sparsity, and further research is needed to understand it.
-	Secondly, the authors also found that BERT trained with a larger amount of pre-training data tends to have a better ability to condense knowledge in relatively fewer parameters. This finding is counter-intuitive because one would expect that increasing the amount of pre-training data would lead to an increase in the number of parameters required to capture the additional information.
-	Thirdly, the authors found that self-supervised learning (SSL) objectives trigger stronger emergent sparsification properties than supervised learning (SL). This finding is also intriguing because one would expect that supervised learning, which provides more explicit information to the model, would lead to better knowledge condensation.

**Weaknesses:**

-	One main downside of this paper is that the paper did not report any result on hardware-friendly sparsity such as N:M, nor it discussed any GPU run time benefit from the induced essential sparsity (if any). While one can understand the study is mainly conceptual, just like the original LTH, I believe the real hardware support is especially important for LLM pruning/sparsity research due to their exploding costs
-	More baselines are desired – currently only LTH is reported. Would essential sparsity meaningfully outperform the simplest baseline of random pruning? How it compares with the pruning result of SparseGPT [18]?
-	In both SparseGPT [18] and Sparsity-May-Cry [75], it was found that larger LLMs are harder to prune. This paper seems to pinpoint the opposite conclusion. Could the authors elaborate why their conclusions seem to contradict [18,75], or not?
-	Sparsity has more benefits beyond efficiency, such as few-shot transfer, robustness to noisy label or other distribution shifts. Many of those were previously demonstrated under LTH or dynamic sparse training settings. I will be curious to see if essential sparsity will own the same merits too.

**Questions:**

Please see the Weaknesses

**Limitations:**

Not discussing more hardware-friendly sparse patterns

---

> ### Author Rebuttal · Authors · 2023-08-10
>
> Many thanks for identifying the significance of our work and finding it promising and practical considering the massive scale of the recent large-scale models. We additionally appreciate that you found that our experiments are quite insightful, have surprising findings, and counter-intutive observations and can lead to opening up several research topics (phase transition etc.). To further address some weaknesses pointed out by you, we would like to address them point-by-point as below:
>
> **1. No results for hardware-friendly sparsity such as N:M:** Thank you for bringing up this point and we are glad to update you that we have additionally **explored fine-grained N:M structured sparsity** (https://arxiv.org/abs/2102.04010) (including widely accepted 2:4 sparsity pattern with real hardware acceleration) which suggests each contiguous block of M values, N values must be non-zero. To our favor, we found that **essential sparsity still holds for N:M sparsity** which can also be identified in a *training-free and data-free manner at FREE COST* bringing actual acceleration for large transformers. We have included **our results on N:M sparsity in Figure 2 of the rebuttal pdf**.
>
> **2. Large models are harder to prune?**
>
> We would like to clarify the confusion that Sparsity may cry [75] states that *on hard tasks (arithmatic reasoning, protein stability etc.), even large models can not be effectively pruned to high sparsity*. It doesn’t mean that the larger the model becomes, the harder it is to prune them. Instead, they claim that the more challenging the task, the more difficult it becomes to prune the model.  Our work observations align with [75] without any contradiction, where we state that the essential sparsity range is dependent on the task complexity (line 193-197 of submitted pdf).
>
> Similarly for SparseGPT [18], they mention “*In general, there is a clear trend of larger models being easier to sparsify, which we speculate is due to overparametrization (paragraph 3 section 4.1)*” which have **no contradiction with our claims**. Moreover, another piece of evidence from Figure 2 from SparseGPT -  "*One key positive finding, illustrated in Figure 2, is that larger models are more compressible: they drop significantly less accuracy at a fixed sparsity, relative to their smaller counterparts*" which doesn't contradict our observation.
>
> **3. More baselines are required like random pruning?** Thank you very much for raising this point and we completely agree that comparison with Random pruning is important. We have **uploaded new results of random pruning, random erk pruning in Figure 1 of the rebuttal pdf**. Note that random pruning performs significantly bad wrt. our one-shot magnitude pruning approach. We promise to include these results in the final version.
>
> In addition to your interest, **Figure 3(a) in our rebuttal draft** illustrates that our essential sparsity observations hold **true even for modern LLMs (Vicuna-B)**, sending a favorable signal about the hidden existence of high-quality sparse subnetwork which can be identified at free in dense pre-trained checkpoints.
>
> To further enrich our study, we replaced OMP with the recently proposed SparseGPT and found it to have generally consistent trends with OMP (**Figure 3(b) in our rebuttal draft**). In addition, it is interesting to observe that better-designed pruning strategies such as SparseGPT can further push the boundary of essential sparsity and identify better sparse subnetworks at comparatively higher sparsity ratios: yet at higher compute costs. We leave as future work on how to close the performance gap between OMP and SparseGPT.
>
> **4. Sparsity beyond Efficiency?** Thank you for bringing it up and we agree that sparsity has benefits beyond efficiency (eg. robustness, few-shot, etc). Unfortunately due to limited time for rebuttal, we leave this experiment for future (camera-ready) as it is not directly related to the primary scope of this work.
>
> We sincerely hope our responses have clarified many of your concerns, and please do not hesitate to let us know what else we could do in order to convince you of a rating upgrade.

---

> > ### Comment · Reviewer_dYgU · 2023-08-20
> >
> > Thank you so much for the reply! I have read other reviews too. It appears that the authors have addressed all concerns properly.
> >
> > I think the paper has presented more-than-sufficient insight and back-up results to warrant its acceptance. N:M sparsity and Vicuna-7B results are even nicer additions. In particular, beyond LLM, this paper reports ViT results too, as well as pre-training dynamics from scratch (which were not observed in peer LLM pruning works).
> >
> > I also don't feel it necessary nor reasonable, to ask for some colossal model (LLaMA 65B, Bloom 175B) done within rebuttal time window.
> >
> > I am raising my score to 8 to champion this solid work.

---

> > > ### Author Response · Authors · 2023-08-20
> > > **Author Response to dYgU**
> > >
> > > We are extremely glad that you find our work solid. We deeply appreciate and thank you for your strong support for the work and identifying its merits. We’re particularly grateful for your agreement on the sufficiency of our current/newly added experiments, and the impracticality of running ‘colossal models’ within a few days.

---

### Official Review · Reviewer_QH66 · 2023-07-06

**Soundness:** 2 fair
**Presentation:** 2 fair
**Contribution:** 3 good
**Rating:** 6
**Confidence:** 3

**Summary:**

This research paper focuses on the following notion for large pre-trained models: "essential sparsity", the idea that a sharp drop in fine-tuning performance occurs after one-shot pruning relative to the level of sparsity. The authors propose that large and overparameterized models can be pruned without additional computational expense, showing this to be true across a range of tasks in both computer vision and natural language processing. Additionally, the authors found an interesting occurrence of abrupt sparsification during pre-training, indicating that models trained with larger datasets tend to achieve knowledge abstraction with fewer parameters. Their findings also revealed that self-supervised learning objectives tend to trigger stronger emergent sparsification properties than supervised learning. They argue that identifying and understanding these inherent high-quality sparse patterns could make fine-tuning large models more practical and environmentally friendly.

**Strengths:**

The paper makes a number of interesting observations about the notion of essential sparsity, its relation to "winning ticket" networks (see LTH), the idea that better knowledge abstraction may be feasible with fewer parameters (as long as we train with more data), and that self-supervised learning tends to have better emergent sparsification than supervised learning.


**Weaknesses:**

Unfortunately, these interesting observation mentioned in the Strengths section, are also too "empirical" or anecdotally shown in the paper. It is hard to be convinced that these are general results -- and they are not substantiated with any form of of theory or at least some explanations/justifications based on results from other papers.

In general, I would describe this paper as "it is moving in a good direction but it is not ready for publication yet" -- at least not at the top-tier conference of deep learning.

**Questions:**

There is a completely different approach to get the benefits of essential sparsity -- namely to start with and train a sparse network. That would be even less computationally intensive than training a dense network and then doing OMP on that.
Are the authors familiar with methods such as SynFlow, PHEW, Synflow++, etc that do pruning before training? If not, I suggest that they also consider those and compare the performance they get from OMP versus those networks.

 The two more interesting observations of the paper (the idea that better knowledge abstraction may be feasible with fewer parameters (as long as we train with more data), and that self-supervised learning tends to have better emergent sparsification than supervised learning) are presented very briefly, without a sufficiently deep analysis in my opinion.

**Limitations:**

One limitation of course is that the claims of the paper are not substantiated in the case of really large foundation models used today -- but of course it is hard for academic researchers to experiment with the training of such models.

Another issue is that the paper does not explicitly state the limitations of the proposed method. For example, in the end of section 6 there are some rather "hidden statements" about the benefits of LTH and IMP, which is a very important point.

---

> ### Author Rebuttal · Authors · 2023-08-10
>
> We would like to thank you for your time to review our work.  We would like to address all the weaknesses pointed out by you point-by-point below:
>
> **1. Experiments are too experimental and hard to be convinced that these are general results:** We would like to highlight that sparse neural networks are new frontier for deep learning theory, and limited theoretical works are available to augment experimental observations and algorithm designs.
> * Although we are heavily empirical, we would like to bring your attention to recent theoretical works  in deep learning theory [https://arxiv.org/abs/2112.11027 , https://arxiv.org/abs/1909.05122 , https://arxiv.org/abs/2002.09277, https://arxiv.org/abs/1903.09367 ] which use the sparsity modeling to understand the implicit regulaization impact and over-parameterization of DNN with growing scale supporting our empirical observations. We promise to cite them in our final version to provide enough theoretical support for our observations.
> * About supervised vs self-supervised findings, our results are again consistent with prior work [https://arxiv.org/abs/2012.06908] on small-scale models, illustrating an important signal that self-supervised learning is more sparsity friendly. We conjecture that sparsity is one of the key structure prior for unsupervised learning [https://openreview.net/pdf?id=TJ2nxciYCk-, https://arxiv.org/pdf/2207.04630.pdf ]; and while we do not have a theoretical explanation yet (leaving it for future work), we believe that self-sup learning inherently induces better sparse patterns during training.
> * To further augment that our finding can be generalized to even modern-day large-scale models, we have attached **additional experiments with Vicuna-7B in the rebuttal pdf Figure 3**. The need to find a high-quality sparse subnetwork in a training-free and data-free approach is significantly important considering the exploding size of LLMs, where the conventional iterative prune-retrain strategy becomes impractical.
> * We have also attached **new results exploring fine-grained N:M structured sparsity** (https://arxiv.org/abs/2102.04010) (including widely accepted 2:4 sparsity pattern with real hardware acceleration) which suggests each contiguous block of M values, N values must be non-zero. To our favor, we found that essential sparsity still holds for N:M sparsity which can also be identified in a **training-free and data-free manner at FREE COST** bringing actual acceleration for large transformers. Please check **Figure 2 of the rebuttal pdf**.
>
>
> **2. Less computationally intensive way: start with a sparse network and perform training?**
>
> We apologize for the confusion, and we would like to highlight that **our work focuses on identifying the sparse patterns in pre-trained models**. Note that we perform OMP on a pre-trained checkpoint and this is *a training-free and data-free approach* (cheapest possible wrt. SynFlow or SynFlow++: Yes we are very familiar with Synflow) to sparsify the pre-trained checkpoint. Note that, first, we are not doing pre-training but simply pruning pre-trained models in resource-constrained environments to identify the sparse subnetwork for free.
>
> Secondly, during fine-tuning of the identified sparse subnetwork, we only do sparse training (as you suggested). We want to clarify that like OMP,  SynFlow can be smoothly integrated with our settings. To your interest, *we applied Synflow and found that Synflow barely outperforms our extremely cheap approach* as shown in the **Figure 1 of our rebuttal pdf**.
>
> **KEY POINT** of this work is not to demonstrate sparsity, BUT it is to demonstrate *how easy this good sparsity could be achieved in large model at no cost* due to emergent behaviors. We are not competing with LTH or any other fancy pruning work, but we are interested in whether it is necessary to LTH or other expensive methods within the essential sparsity range.
>
> **3.Claims of the paper are substantiated for really large foundational models used today?** Thank you for bringing up this point and we provide **additional experiments with Vicuna-7B on popular MMLU benchmark in the rebuttal pdf Figure 3**. Based on our results in Figure 3, it is interesting to observe that our essential sparsity observations hold true even for modern LLMs, sending a favorable signal about the hidden existence of high-quality sparse subnetwork which can be identified at free in dense pre-trained checkpoints. We also provide new results which extend our observations for fine-grained N:M structured sparsity with real hardware potential.
>
> **4. Limitations are not explicitly outlined?** We really appreciate your concerns and we promise to add a separate limitation section in the paper explicitly mentioning the limitations of our work (such as the benefits of IMP in high sparsity regime, theoretical evidence, scaling up to 10B+ model parameters, etc.).
>
> We sincerely hope our responses have clarified many of your concerns, and please do not hesitate to let us know what else we could do in order to convince you of a rating upgrade.

---

> > ### Comment · Reviewer_QH66 · 2023-08-10
> >
> > Thank you for generating some new results to address one of my comments.  I appreciate that and I will increase my score from Borderline to Weak Accept (only considering the comments posted by the other reviews).

---

### Official Review · Reviewer_Vvp4 · 2023-07-07

**Soundness:** 3 good
**Presentation:** 3 good
**Contribution:** 3 good
**Rating:** 4
**Confidence:** 5

**Summary:**

This paper defines essential sparsity and conducts various experiments to analyze the sparsity property of pre-trained models.

**Strengths:**

1. This paper investigates the potential of directly sparsifying the pre-trained models
2. CV and NLP models are both explored.

**Weaknesses:**

1. Evidence on large models is lacking.
2. Some conclusions are not new, e.g., sharp dropping point.

**Questions:**

What about the conclusions on large models like LLaMA 65B or Bloom 175B?

**Limitations:**

The insights brought from this paper is limited.

---

> ### Author Rebuttal · Authors · 2023-08-10
>
> We would like to thank you for your time to review our work. We appreciate your feedback and would like to address the concerns you raised regarding our work's weaknesses. However, we believe that there might be some key contributions that may not have been fully acknowledged in your assessment. We kindly suggest taking into consideration the viewpoints of other reviewers as well, as they might provide a more comprehensive perspective on our work.
>
> 1. To address your concerns related to evidence on large-scale models, we have included **new experiments on Vicuna-7B in the rebuttal pdf Figure 3** (note we have OPT-1.3B results in the submitted draft) which aligns with our observations in the paper. Secondly, we would clarify that although the abrupt drop phenomenon is observed previously; we would like to highlight that our key point is to study this observation in a multi-dimensional way wrt training strategy, pruning strategy, model-scale, data modality, and dataset size. Not just extending to pre-trained models, our multi-dimensional study reveals many interesting findings such as sparsity ratios below the abrupt drop marker are agnostic to pruning strategies (iterative vs OMP) and you do not require any fancy and expensive method to identify high-quality sparse subnetworks. At a glance, it might look simple, but we believe this has a huge practical implication considering the impracticality to perform iterative prune and retrain strategy with modern scale LLMs.
>
> Finally, we kindly hope you understand that the request for conclusions on LLaMa 65B and Bloom 175B couldn’t be afforded without large-scale industry-scale hardware.
>
> 2. **Conclusions are not new?** We respectfully disagree. As nicely summarized by other reviewers: **(a) Reviewer XMXq:**  *“...this paper opens up several research topics to be studied. I believe they will be impactful…”* **(b) Reviewer Azn4:** *“...comparison of self-supervised vs fully supervised models is interesting…”* **(c) Reviewer QH66:** *“...paper makes a number of interesting observations about the notion of essential sparsity…”* **(d) Reviewer dYgU:** *“...the authors present several quite surprising findings related to the existence of essential sparsity…”*. Almost all other reviewers found that our empirical observations have wide practical importance to find and utilize the pre-existing sparsity of transformers.
>
> Additionally, we believe our observations related to abrupt sparsification during the pre-training process, supervised vs self-supervised, etc also open exploration playground to theoretically understand and design efficient pre-training strategies for researchers.
>
> We sincerely hope our responses have clarified many of your concerns, and please do not hesitate to let us know what else we could do in order to convince you of a rating upgrade.

---

> > ### Author Response · Authors · 2023-08-15
> > **Author response to Vvp4**
> >
> > Dear Reviewer Vvp4,
> >
> > We thank you for your time to review our work and your constructive comments to improve it, and we really hope to have a further discussion with you to see if our response solves your concerns. We have replied to the important points raised by you such as novelty concerns, limited experiments for large-scale models etc., in our rebuttal response. Since the author-reviewer discussion period has started for a few days, we will appreciate it if you could check our response to your review comments soon. This way, if you have further questions and comments, we can still reply before the author-reviewer discussion period ends.
> >
> > If our response resolves your concerns, we kindly ask you to consider raising the rating of our work. We again thank you for your time and efforts.
> >
> > Best, Authors

---

> > > ### Author Response · Authors · 2023-08-18
> > > **2nd Reminder on feedback**
> > >
> > > Dear Reviewer Vvp4:
> > >
> > > This is our 2nd reminder for you to please read our rebuttal and hopefully update your opinion. As the deadline for the discussion period is approaching, we would appreciate it if you could kindly let us know whether any further questions remain. We’re very confident that our rebuttal responses will address your concerns.
> > >
> > > We would again like to highlight that we have addressed your concerns related to large models and additionally provided more experimental results related to structured N:M sparsity patterns, which significantly increase the value of our work.
> > >
> > > Authors,

---

> > ### Comment · Reviewer_Vvp4 · 2023-08-20
> >
> > 1. Thank the authors for adding Vicuna-7B experiments. However, we think it can not be defined as very large pre-trained models.
> > 2. As for the shart drop point, we can refer to the paper such as https://arxiv.org/pdf/2301.00774.pdf. In figure 1, it has revealed an obvious drop from certain sparsity rates.

---

> > > ### Author Response · Authors · 2023-08-20
> > > **Response to Comment by Reviewer Vvp4**
> > >
> > > We would like to thank you for your time to read our rebuttal and responding back.
> > >
> > > **1. Vicuna-7B is not considered as very large pre-trained model:** We value your concern BUT we would again like to emphasize that 65B or 175B experiments are seriously impractical without the availability of big industry-scale hardware support. Considering your response came very close to the deadline of the discussion period (~1 day before it ends), we would not be able to secure the required hardware support and complete the experiments within the deadline. However, we promise to scale our findings to 65B in the final version of our paper.
> > >
> > > In addition, we are running experiments for Vicuna-13B and it is expected to complete within the next 12 hours and we will update our results as soon as we get it. We sincerely hope, you can understand the hardware constraints and several concurrent works (eg. LLM pruner https://arxiv.org/abs/2305.11627) also restrict their findings to 7-10B scale models.
> > >
> > > **2. Sharp-drop point behavior:** Thank you for raising this concern, and we certainly agree that *almost all pruning methods will observe their performance drop after a certain sparsity level - there is not any surprise nor our main finding*. Unfortunately, it seems **you have missed the key message of our work**.
> > >
> > > * The primary goal of this paper is to show the easiest pruning option, *one shot, magtitude-based, training-free, and data-free pruning* (different from SparseGPT who also requires calibration data and the more costly Hessian estimation) echoing exactly the same behavior as any sophisticated method like LTH within a sparsity range. **The most important finding, as overlooked as prior work,  is summarized** as: *within the induced sparsity range induced by “essential sparsity”, the simplest possible pruning technique as aforementioned performs the same good as any fancy technique like LTH. and even their identified sparse masks are extremely similar*. **This is NEVER revealed by any other work, and it seems other reviewers have appreciated this main merit.**
> > >
> > > * Orthogonal to SparseGPT or any other pruning method: it for first time we reveals how **“easy”** large model pruning is. At least within certain sparsity range, one need not look beyond the *simplest one shot, magtitude-based, training-free, and data-free pruning* - that both presents a strong baseline for future pruning and reveals a strong “in-situ” pruning option (e.g. pruning is as simple as mag sorting). Hence our finding comes with profound practical value too, e.g., for cheap “on-the-fly” LLM pruning at test time adaptive to varying resource availability.
> > >
> > > * We provide some surprising and counter-intuitive findings related to **emerging abrupt sparsification of BERT during pre-training, sparsity dynamics in supervised vs self-supervised settings, first-time controlled pre-training data experiments** which illustrate more data make models more sparser, etc. We also included ViT experiments. All other reviewers have appreciated the significance of our interesting findings and the solidness of our thorough experiments.
> > >
> > > We reiterate to include Vicuna-13B experiments within next 12 hours, and sincerely hope that you will give a look at it before making your final decision. We hope for the best and still open to clarify any doubts if you have to convince you for a rating upgrade before the clock ticks out.

---

> > > > ### Author Response · Authors · 2023-08-21
> > > > **Response to Comment by Reviewer Vvp4**
> > > >
> > > > As per our promise, we additionally provide the performance of Vicuna-13B on the MMLU benchmark within the essential sparsity range (we found it to be ~35\% for Vicuna-13B). Note that MMLU is a comparatively challenging task and we have also provided a comparison with SparseGPT in the following table.
> > > >
> > > > |Sparsity Ratio | Dense | 5% | 10% | 15% | 20% | 25% | 30% | 35% | 40%|
> > > > | ------------- |:-------------:|:-------------:|:-------------:|:-------------:|-------------:|-------------:|-------------:|-------------:|-------------:|
> > > > |One-shot MAG| 39.2 | 39.4| 39.1| 39.30| 39.11| 38.86| 38.7| 37.5| 34.80|
> > > > |SparseGPT| 39.2| 39.5| 39.30| 39.2| 38.75| 38.54| 37.98| 37.6| 36.2|
> > > >
> > > > It can be clearly observed that within the essential sparsity range, both one-shot and SparseGPT have comparable performance and SparseGPT doesn't bring any additional benefit even being computationally much expensive and dependent on calibration data. Removal of ~35% from Vicuna-13B without any cost again aligns with the key message of our work.
> > > >
> > > > We sincerely hope that these new results will further help you in making the final decision. We will continue our experiments for LLama-75 and BLOOM-175B and update results in the final version. But we want to **emphasize again that our original submission + all the additional results are already strong and solid for the reasons we mentioned, and we believe more evaluations are only going to help support our work.**
> > > >
> > > > Best,
> > > > Authors

---

### Official Review · Reviewer_AZn4 · 2023-07-27

**Soundness:** 2 fair
**Presentation:** 2 fair
**Contribution:** 2 fair
**Rating:** 4
**Confidence:** 3

**Summary:**

The paper postulates existence of "Essential Sparsity" in large pre-trained transformer models. The "Essential Sparsity" is defined by the paper as a sparsity threshold beyond which further pruning of weights leads to a large performance drop. It considers (1) one-shot pruning and (2) lottery ticket pruning of networks with fine-tuning. Experiments are conducted on both large language and vision transformer models.

**Strengths:**

While the sparsity has been studied a lot in the neural network literature, this work studies the sparsity for pre-trained language and vision models. Given the wide adoption of these networks, the study is important. The paper covers a number of experiments for large vision as well as language models. The comparison of self-supervised vs fully supervised models is interesting.

**Weaknesses:**

1. Existence of sparsity in transformers in both language [a,b] and vision [c] has been discussed in the prior literature. The drop of performance in networks beyond certain threshold has also been covered in the literature. I do believe the current work is the first one to formally define it, however it will be good if the work can put the contributions in proper context.

2. Many of the "surprising" findings of the work appear to be over-claimed. For instance,

    i. The abrupt drop in the performance of networks beyond certain sparsity thresholds is shown by the original LTH paper and the current work extends it to pre-trained network.

    ii. The similarity of sparse masks for various downstream tasks is also not very surprising given that the networks was pre-trained on a large massive dataset and the downstream tasks comprise of very small similar sets of data (often even the subset of pre-training data).

    iii. L50 (and L66) says that "a significant proportion of the weights in them can be removed for free" while it is hardly free given the network has been heavily pre-trained.

3. The mathematical definition of Essential Sparsity is somewhat unclear (what exactly does $1-\frac{||m||_0}{|m|}$ represent?). Also calling it essential for the network is a bit misleading. I agree dropping it would hurt the performance of the network but it is not "essential" for the network to perform well. A fully dense network usually performs the best.

4. Minor fix - Labels for x-axis is missing in most of the figures.

References:

a. Chen, Tianlong, et al. "The lottery ticket hypothesis for pre-trained bert networks." NeurIPS 2020

b. Dettmers, Tim, et al. "Llm. int8 (): 8-bit matrix multiplication for transformers at scale." NeurIPS 2022

c. Girish, Sharath, et al. "The lottery ticket hypothesis for object recognition." CVPR. 2021.

**Questions:**

While some of the analyses of the paper is interesting, I would appreciate authors comments on above weaknesses.

**Limitations:**

The paper doesn't cover discuss the limitations of the study in details. For example, while the sparsity and efficiency of models both during training and inference is an important problem, it should be kept in mind that unstructured sparsity alone is not good enough to obtain speed gains. There is no discussion of potential speed gains with the unstructured sparsity (this doesn't necessarily mean FLOPs but speed improvement during inference/training), or training time of different networks, etc.

---

> ### Author Rebuttal · Authors · 2023-08-10
>
> We would like to thank you for your time to review our work and glad that you find our work important and that some of our observations are very interesting. We would like to address all the weaknesses pointed out by you point-by-point below:
>
> **1. Existence of sparsity in transformers in both language and vision has been discussed in the prior literature?** Thank you for outlining your concern. We would like to emphasize that although there exist several works discussing the existence of sparsity in transformers:
> * Our work primarily focuses to bring attention to the pre-existing sparse patterns in a **TRAINING-FREE** and **TASK-FREE** setting which is critically important considering the exploding scale of transformers. As pointed out by reviewers XMXq and dYgU, our observations provide a promising and practically efficient approach to finding and utilizing the pre-existing sparsity of transformers and potentially open up discussion around many research topics (eg. phase transition, etc).
> * In addition, we are the first to scale up our observations for sparsity **at billion-level transformers** (additional results in rebuttal pdf **include Vicuna 7B results - check Figure 3**) and illustrate that you do not require any iterative pruning and retraining in the essential sparsity range. We for the first time provide empirical evidence that sparse masks obtained by expensive iterative prune and retrain procedures and simple OMP without any retraining are significantly similar indicating LTH is not doing anything magical in essential sparsity regimes and OMP can identify matching subnetworks.
> * Lastly, our observations related to abrupt sparsification during the pre-training process, supervised vs self-supervised, etc also open exploration playgrounds to theoretically understand and design efficient pre-training strategies for researchers.
>
> **NOTE:** We also include new results on Vicuna-7B to illustrate the existence of essential sparsity within them; indicating a sound message that there potentially **exists an easy way to sparsify modern-day LLMs without access to any training data, no retraining, in one-shot fashion** before observing abrupt performance drop.
>
> **KEY POINT** of this work is not to demonstrate sparsity, **BUT** it is to demonstrate how easily this good sparsity could be achieved in the large model at no cost. We are not competing with LTH or any other pruning work, but we are interested in whether it is necessary to use LTH or other expensive methods.
>
> **2. Over-claimed observations:**
> * **Abrupt drop is observed by original LTH paper:** We would clarify that although an abrupt drop in the performance of networks beyond certain sparsity thresholds is shown by the original LTH paper; our key point is to study this observation in a multi-dimensional way wrt. training strategy, pruning strategy, data modality, and model-scale. Not just extending to pre-trained models, our multi-dimensional study reveals many interesting findings such as sparsity ratios below the abrupt drop marker are agnostic to pruning strategies (iterative vs OMP) and you do not require any fancy and expensive method to identify high-quality sparse subnetworks. At a glance, it might look simple, but this has a huge practical implication considering the impracticality to perform iterative prune and retrain strategy with modern scale LLMs.
> * **Similarity of sparse mask for various downstream tasks is not interesting:** We would like to highlight that unlike developing downstream task-dependent sparse mask (https://arxiv.org/pdf/2303.14409.pdf, https://arxiv.org/abs/2012.06908, etc), our work again brings focus on the key question: *Is always required to have a task-dependent mask which only works for single task considering the computational cost involved in identifying them?* Our observations find that within the essential sparsity range, it is not required to search for a task-dependent mask and a cheap one-shot mask is as good as an expensive task-dependent mask identified by IMP of LTH.
> * **Our argument of free is hardly free given the network has been heavily pre-trained?** We apologize for the confusion, and we think there is a misunderstanding of the term “for free” in the wrong context. When we say “for free”, given a pre-trained model, with our OMP settings you do not need an additional computational budget to iteratively prune and retrain the model weights to identify sparse subnetworks (impractical for modern LLMs without industry-standard hardware), thereby no pruning overhead.
>
> **3. Limitation [unstructured sparsity is not good enough]:** Thank you for bringing up this point and we are glad to update you that we have **additionally explored fine-grained N:M structured sparsity** (https://arxiv.org/abs/2102.04010) (including widely accepted 2:4 sparsity pattern with real hardware acceleration) which suggests each contiguous block of M values, N values must be non-zero. To our favor, we found that **essential sparsity still holds for N:M sparsity** which can also be identified in a training-free and data-free manner at FREE COST bringing actual acceleration for large transformers. We have included our results on N:M sparsity in **Figure 2 of the rebuttal pdf**.
>
> **4. Essential for network is misleading?** We would like to clarify the confusion. When we say essential sparsity it does not mean it is essential for the network. However, it means that the network reveals robustness to sparsification and its performance doesn’t significantly drop wrt. pruning with the essential sparsity range. We follow previous works (https://arxiv.org/abs/1901.09181, https://arxiv.org/abs/1902.09574, https://arxiv.org/abs/2101.09048) and define the model sparsity as  $1-\frac{\|m\|_0}{|m|}$, where $|m|_0$ is the number of non-zero weights and $|m|$ refers to the total number of weights.
>
> Hope our rebuttal clarifies all your doubts and please let us know what else we could do to convince you of a rating upgrade.

---

> > ### Author Response · Authors · 2023-08-15
> > **Author Response to Reviewer AZn4**
> >
> > Dear Reviewer AZn4,
> >
> > We thank you for your time to review our work and your constructive comments to improve it, and we really hope to have a further discussion with you to see if our response solves your concerns. We have replied to the important points raised by you such as novelty concerns, limited gain from unstructured sparsity, over-claimed observations, clarification regarding the definition of essential sparsity etc., in our rebuttal response.
> >
> > To augment our rebuttal response, we would like to further add more clarification related to novelty concerns of abrupt drop in our work. We certainly agree that *almost all pruning methods will observe their performance drop after a certain sparsity level - there is not any surprise nor our main finding*. We would like to highlight the key message as follows:
> >
> > * The primary goal of this paper is to show the easiest pruning option, *one-shot, magnitude-based, training-free, and data-free pruning*  echoing exactly the same behavior as any sophisticated method like LTH within a sparsity range. **The most important finding, as overlooked as prior work,  is summarized** as: *within the induced sparsity range induced by “essential sparsity”, the simplest possible pruning technique as aforementioned performs the same good as any fancy technique like LTH. and even their identified sparse masks are extremely similar*. **This is NEVER revealed by any other work, and it seems other reviewers have appreciated this main merit.**
> >
> > * For first time we reveal how **“easy”** large model pruning is. At least within a certain sparsity range, one need not look beyond the *simplest one shot, magnitude-based, training-free, and data-free pruning* - that both presents a strong baseline for future pruning and reveals a strong “in-situ” pruning option (e.g. pruning is as simple as mag sorting). Hence our finding comes with profound practical value too, e.g., for cheap “on-the-fly” LLM pruning at test time adaptive to varying resource availability.
> >
> > * We provide some surprising and counter-intuitive findings related to **emerging abrupt sparsification of BERT during pre-training, sparsity dynamics in supervised vs self-supervised settings, first-time controlled pre-training data experiments** which illustrate more data make models more sparser, etc. We also included ViT experiments. All other reviewers have appreciated the significance of our interesting findings and the solidness of our thorough experiments.
> >
> > Since the author-reviewer discussion period has started for a few days, we will appreciate it if you could check our response to your review comments soon. This way, if you have further questions and comments, we can still reply before the author-reviewer discussion period ends.
> >
> > If our response resolves your concerns, we kindly ask you to consider raising the rating of our work. We again thank you for your time and efforts.
> >
> > Best, Authors

---

> > > ### Author Response · Authors · 2023-08-18
> > > **2nd Reminder on feedback**
> > >
> > > Dear Reviewer AZn4:
> > >
> > > This is our 2nd reminder for you to please read our rebuttal and hopefully update your opinion. As the deadline for the discussion period is approaching, we would appreciate it if you could kindly let us know whether any further questions remain. We’re very confident that our rebuttal responses will address your concerns.
> > >
> > > We would again like to highlight that we have addressed your concerns related to novelty, limited gain from unstructured sparsity, over-claimed observations, clarification regarding the definition of essential sparsity etc., and additionally provided more experimental results related to structured N:M sparsity patterns, which significantly increase the value of our work.
> > >
> > > Authors,

---

> > > ### Comment · Reviewer_AZn4 · 2023-08-21
> > >
> > > Thank you for your response (and apologies for late reply). I've gone through the rest of the reviews and author responses.
> > >
> > > - Regarding mine (and other reviewers') concern on efficacy of unstructured sparsity - authors have provided some additional experiments on 2:4 sparsity patterns (although it appears that performance of model drops a bit on the downstream tasks and it is still unclear how much is the improvement in inference speed with the achieved 2:4 sparsity)
> > > - "We are not competing with LTH or any other pruning work, but we are interested in whether it is necessary to use LTH or other expensive methods" - IMHO given the repeated claims that the one-shot pruning works as well (or nearly as well) as LTH, the work is competing with pruning works such as LTH
> > > - Regarding my concerns on overclaims - the authors continue to call it a data-free, training-free pruning method which I believe is misleading for the readers.
> > >
> > > Overall my main concern is **not** the experiments on big models but presenting the paper in a fair manner and providing an essential analysis or trade-offs between one-shot pruning vs other pruning methods (say IMP). There are obvious advantages in terms of less training time and such. But this proper trade-off between training time / test-time accuracy / speed improvements is not properly revealed in the paper.
> > >
> > > I am currently keeping my score in a hope that authors would significantly tone down their claims and provide a fair assessment and limitations of the work.

---

> > > > ### Author Response · Authors · 2023-08-21
> > > > **We would happily adjust the tone - but hope for more clarifications**
> > > >
> > > > Dear Reviewer AZn4,
> > > >
> > > > Despite last-minute, we are delighted to hear further from you and are thankful for your time. **TL;DR; we are happy to adjust our tone and to "provide a fair assessment and limitations of the work" and we proposed a few actions below, although we would appreciate if you could also clarify some concerns a bit more**. From what we understand, all concerns are caused by confusion or miscommunication, and shall be easily fixable in current writing.
> > > >
> > > > We would be very thankful if you could take another moment to review our (as timely as possible) reply, and we will keep standing by from now on, if you want to discuss more before the rebuttal closes in ~8 hours.
> > > >
> > > > (1) 2:4 sparsity is the standard convention of "structured sparsity" that is evaluated by current few LLM pruning methods such as sparseGPT and Wanda (available later than NeurIPS deadline); and is a commonly accepted practice in GPU model acceleration. We have reported other N:M forms too for completeness. We will happily add real GPU hardware inference time in the final version; meanwhile our reporting has followed the standard convention in prior/peer works (e.g., neither sparseGPT nor Wanda reported hardware time of their N:M models in latest arxiv versions)
> > > >
> > > > (2)
> > > >
> > > > "The work is competing with pruning works such as LTH" - sorry for the confusion. **We never have intention to hide or avoid comparing with LTH, and we re-state our comparison rationale in bullets below**:
> > > > - Essential sparisty has two main characterization points: (i) within this range, the one-shot pruned model performs as well as or better than dense baseline; (ii) beyond this range, the sudden accuracy drop is observed
> > > > - LTH at best preserves the dense model accuracy (or very slightly better), while our method achieves the same behavior within essential sparsity range. Therefore, at least within essential sparsity range, ours is comparable to LTH in accuracy (through the *indirect, yet sound* comparison using dense model as "anchor")
> > > > - Meanwhile, ours has "obvious advantages in terms of less training time and such", which is our main point in this paper: one can achieve good/best pruning without paying the expensive cost, within the essential sparsity range
> > > > - Accurately performing LTH/IMP on billion-level LLM is hardly practical for most (if not all) research teams. Despite so, we tried my best to report *direct* comparison with LTH and ours on bert-base and ViT-base in Section 6. While that section is currently focused on reporting mask similarity not accuracy (due to the first bullet), we do have all "accuracy" results of LTH for Section 6 experiments and we can add them all if the reviewer feels helpful
> > > >
> > > > We sincerely hope the above has further clarified and solidified our "repeated claims that the one-shot pruning works as well (or nearly as well) as LTH".
> > > >
> > > > (3)
> > > >
> > > > "data-free, training-free pruning method"  - we suggest you perhaps **unintentionally misunderstand our terms in this context**. This claim is made w.r.t the context of existing LLM pruning methods. We hereby clarify more below, and we can integrate the clarifications in final paper to avoid any future reader confusion:
> > > > - **Data free**: this is commented w.r.t. sparseGPT or Wanda  (available later than NeurIPS deadline), both of which will use a small set of "calibration data" to run inference and compute activations (because they need activation values as part of their pruning criteria, thus dependent on input). Our method relies on nothing but pre-trained weight value, hence no such need. We believe our "no activation/inference" criteria is very helpful when pruning needs to be performed on a real-time, or resource-adaptive basis. To help clarify, we are willing to revise "**Data free**" into "**Inference free**": please let us know if that sounds clearer to you
> > > > - **Training free**: this is commented w.r.t. sparseGPT, which relies on solving layer-wise Hessian-inspired optimization over pre-trained weights. Ours belongs to the same camp as Wanda, with no need of post-optimization, which incurs negligible time overhead compared with SparseGPT. (ours is even way cheaper, due to no inference) We apparently never talk about "free of pre-training" overhead or so. To help clarify, we are willing to revise "**Training free**" into "**Post-optimization free**": please let us know if that sounds clearer to you
> > > > - Your mention of "this proper trade-off between training time ..." is confusing us, since indeed our pruning has no overlead related to post-training (except the time used to sort pre-trained weight magnitudes). We fail to comprehend why a post-(pre)training pruning method must be responsible for that "the network has been heavily pre-trained" - this deviates from common evaluation practice for this type of methods
> > > >
> > > > We will always do our part as serious & responsible authors :-) We hope our above responses settle more of your concerns and please don't hesitate to let us know if we could address more!

---

> > > > > ### Comment · Reviewer_AZn4 · 2023-08-21
> > > > >
> > > > > Thank you for your response and it certainly provides more clarity. My current belief is that the paper will greatly benefit from a major writing revision.

---

> > > > > > ### Author Response · Authors · 2023-08-21
> > > > > >
> > > > > > Thank you again for your prompt response and we are glad that our detailed explanation finally provided clarity and it was able to resolve your doubts and concerns.
> > > > > >
> > > > > > Please note that all of the **above-suggested changes are actually easy to incorporate in our submission draft** and we promise to make these modifications in camera-ready.
> > > > > >
> > > > > > Specifically, none of our main technical conclusions nor their associated analysis will be affected - and it seems now we agree that there also exists no "over-claim" after we clarified above.
> > > > > > In our revision, we will clarify a few terms, mainly:
> > > > > > - explaining "for free" (note we didn't use training-free/data-free in the original paper but rather used them to explain "for free" in the previous rebuttal) explicitly as "free of inference or post-optimization"
> > > > > > - explaining our comparison with LTH explicitly as "same as the unpruned baseline performance as well as LTH performance within the essential sparsity range"
> > > > > > - other appropriate tone adjustments, if further suggested by the reviewer explicitly
> > > > > >
> > > > > > etc. Yet other than those additional clarifications (which are very helpful, and we are grateful for you bringing them up), we honestly did not see a "major writing revision" needed.
> > > > > >
> > > > > > We would humbly appreciate it if you consider raising your score based on the technical merits of our work and our rebuttal; which has been key in resolving your doubts.

---

### Official Review · Reviewer_XMXq · 2023-07-29

**Soundness:** 3 good
**Presentation:** 3 good
**Contribution:** 3 good
**Rating:** 7
**Confidence:** 4

**Summary:**

This paper delves into the natural sparsity of large-scale pretrained transformers that could be found without additionally training the models with sparsity optimization targets. The authors find that when simply low-magnitude weights are zeroed out, 1) up to a certain sparsity level,  the pruned transformers do not exhibit performance drop on downstream tasks after fine-tuning. 2) the sparsity of weights abruptly increases at certain training iterations 3) simple magnitude-based pruning shows similar performance as the Lottery Ticket Hypothesis method up to a certain sparsity level. The phenomenon is observed across many different LLM and vision transformer architectures. Also interestingly, self-supervised weights are more sparse than supervised weights.


**Strengths:**

This paper presents interesting findings on the nature of pretrained weights of large-scale models based on substantial amount of experimental grounds. The authors emphasize that the use of essential sparsity to prune weights can be done without repetitive train-prune-retrain routine. Considering the massive scale of recent large-scale transformers, it is a promising and practically efficient approach to find and utilize the sparsity of transformers.

In addition, this paper opens up several research topics to be studied. I believe they will be impactful.
- Sparsity pattern from self-supervised learning and supervised learning
- Sparsity pattern change by training on larger-scale datasets
- Acceleration of large-scale transformers by pruning weights


**Weaknesses:**

- Practical gains after finding the sparsity pattern

In contrast to learning-based pruning methods (such as LTH), the essential sparsity has the raw pattern of weight magnitude. Although it is interesting to find the sparsity is there by nature, as mentioned in L90-91, such patterns can’t be directly used to bring actual acceleration of neural networks. Training-based methods can set optimization goal on sparsity patterns to alter the sparsity pattern to make the neural network accelerable, however, such approach seems to be difficult for essential sparsity. While it is easier to find the sparsity pattern with essential sparsity, what would be the practical gains of the proposed method after finding the sparsity?

- Why does sparsity occur?

The authors present interesting comparison of impact of sparsity on model accuracy on various conditions - model size & architecture, supervised vs self-supervised learning. It is interesting to know of such results, but why does such differences happen? How can we understand the phenomenon?

- typos

L36 pre0trained -> pre-trained


**Questions:**

Please refer to the weaknesses.

**Limitations:**

No negative societal impact is expected.

---

> ### Author Rebuttal · Authors · 2023-08-09
>
> Many thanks for identifying the significance of our work and finding it promising and practical considering the massive scale of the recent large-scale models. We additionally appreciate that you found that our experiments are quite solid and our observations can lead to opening up several research topics. To further address some weaknesses pointed out by you, we would like to address them point-by-point as below:
>
> **1. Limited practical acceleration due to unstructured sparse patterns?** Thank you for bringing up this point and we are glad to update you that we have additionally explored fine-grained N:M structured sparsity (https://arxiv.org/abs/2102.04010) (including widely accepted 2:4 sparsity pattern with real hardware acceleration) which suggests each contiguous block of M values, N values must be non-zero.
>
>  * To the surprise in our favor, we found that essential sparsity **still holds for N:M sparsity** which can also be identified in a training-free and data-free manner at FREE COST bringing actual acceleration for large transformers. We have included our results on N:M sparsity in Figure 2 of the rebuttal pdf.
>
> * We would like to address that although optimization goals on sparsity patterns can be set up during training; it is impractical for this to be adopted to current gigantic transformers without industry-scale hardware.
>
> * On the other hand, the pivotal contribution of the essential sparsity is to bring attention to FREE of COST pre-existing sparse patterns without any training requirements.
>
> **2. Why does sparsity occur?** Thank you for raising this question. Prior literature has consistently observed that higher sparsity comes at a natural consequence of model size. [https://openreview.net/pdf?id=TJ2nxciYCk-] found that large transformers often have unactivated sparse neurons (activation space) and our results can be viewed as a counterpart in weight space. Similarly, [https://cbmm.mit.edu/sites/default/files/publications/Theoretical_Framework__How_Deep_Nets_May_Work_15.pdf] argues that certain deep architectures – such as CNNs and transformers work very well because they significantly exploit the general property of compositional sparsity. In addition, in deep learning theory [https://arxiv.org/abs/2112.11027 , https://arxiv.org/abs/1909.05122 , https://arxiv.org/abs/2002.09277, https://arxiv.org/abs/1903.09367 ] there are many work using the sparsity modeling to understand the implicit regularization impact and over-parameterization of DNN with growing scale. Regarding your point about supervised vs self-supervised, our results are consistent with prior work [https://arxiv.org/abs/2012.06908 ] on small-scale models, illustrating a important signal that self-supervised learning is more sparsity friendly. We conjecture that sparsity is one of the key structure prior for unsupervised learning [https://openreview.net/pdf?id=TJ2nxciYCk-, https://arxiv.org/pdf/2207.04630.pdf ]; and while we do not have a theoretical explanation yet (leaving it for future work), we believe that self-sup learning may inherently induce better sparse patterns during training.
>
> Finally, thank you for pointing out some typos, and we promise to correct them all in the camera ready version along with a more detailed related section discussing the above prior works.
>
> We hope our responses have clarified many of your concerns, and please do not hesitate to let us know what else we could do in order to convince you for a rating upgrade.

---

> > ### Comment · Reviewer_XMXq · 2023-08-22
> >
> > After reading the rebuttal, I still think the essential sparsity to be far from being practical.
> > In rebuttal Figure 2, the authors present performance drops from several different N:M sparsity patterns. However, I don't think any of the other N:M patterns other than 2:4 are valid. Albeit general N:M sparsity is theoretically possible, they require hardware support (such as tensor cores) to have actual acceleration. 2:4 sparsity has been supported by GPU hardware for a few years, however, no other patterns have been supported. Some patterns where n or m are powers of 2 are theoretically explored, however, the patterns in Rebuttal Figure 2 (1:10, 1:6, 2:10) are not practical. (Is it 1:10 or 9:10?)
> > * A. Zhou et al., LEARNING N:M FINE-GRAINED STRUCTURED SPARSE NEURAL NETWORKS FROM SCRATCH, ICLR 2021
> >
> > As 2:4 sparsity is losing accuracy, I don't consider it to be safely accelerable without additional training. Given that the authors could provide more objective comments in the revision, I am ok with it.
> >
> > I still find it an interesting paper. I'm between Accept and Weak Accept.

---

### Author Rebuttal · Authors · 2023-08-10

We would like to thank all the reviewers for their time to review our work and offering important suggestions. In this pdf, we attach some key experiments requested by reviewers which further strengthen the impact of our work. We summarize our results as follows:

* **[Requested by Reviewer QH66 and dYgU] Figure 1** provides additional results related to the performance comparison of SynFlow, and the important baseline of Random Pruning + Random pruning with ERK; and illustrates that SynFlow doesn't bring any additional benefit despite being more expensive than OMP.
* **[Requested by Reviewer XMXq, AZn4, dYgU BUT for all] Figure 2** provides additional results exploring fine-grained N:M structured sparsity (https://arxiv.org/abs/2102.04010) (including widely accepted 2:4 sparsity pattern with real hardware acceleration) and show our essential sparsity observations holds true for N:M sparse patterns.
* **[Requested by Reviewer Vvp4, QH66, dYgU BUT for all] Figure 3** provide additional experiments with *Vicuna-7B on popular MMLU benchmark*. Based on our results in Figure 3, it is interesting to observe that our essential sparsity observations hold true even for modern LLMs, sending a favorable signal about the hidden existence of high-quality sparse subnetwork which can be identified at free in dense pre-trained checkpoints.

In last, we again thank all the reviewers and hope our additional results + rebuttal responses can clarify their doubts. We are more than happy to provide any further explanations required.

Best,
Authors 9125

---

### Decision · Program_Chairs · 2023-09-21

**Decision:**

Accept (poster)

**Comment:**

There was a wide spread in initial ratings. Author rebuttal helped address majority of the concerns, though there are still some lingering questions from the reviewers in the discussion. After reading through the discussion and taking the author rebuttal and reviewer opinion into account, I recommend acceptance. Many of the pending concerns can be addressed in text, and I strongly recommend that authors incorporate these changes to further improve their manuscript.